# LMUFORMER: LOW COMPLEXITY YET POWERFUL SPIKING MODEL WITH LEGENDRE MEMORY UNITS

**Zeyu Liu**[*]**, Gourav Datta**[*,†]**, Anni Li, Peter A. Beerel**
University of Southern California, Los Angeles, CA, USA
`{liuzeyu,gdatta,annili,pabeerel}@usc.edu`
[*]Equally contributing authors   [†]Currently employed at Amazon Inc.

## ABSTRACT

Transformer models have demonstrated high accuracy in numerous applications but have high complexity and lack sequential processing capability making them ill-suited for many streaming applications at the edge where devices are heavily resource-constrained. Thus motivated, many researchers have proposed reformulating the transformer models as RNN modules which modify the self-attention computation with explicit states. However, these approaches often incur significant performance degradation. The ultimate goal is to develop a model that has the following properties: parallel training, streaming and low-cost inference, and state-of-the-art (SOTA) performance. In this paper, we propose a new direction to achieve this goal. We show how architectural modifications to a fully-sequential recurrent model can help push its performance toward Transformer models while retaining its sequential processing capability. Specifically, inspired by the recent success of Legendre Memory Units (LMU) in sequence learning tasks, we propose LMUFormer, which augments the LMU with convolutional patch embedding and convolutional channel mixer. Moreover, we present a spiking version of this architecture, which introduces the benefit of states within the patch embedding and channel mixer modules while simultaneously reducing the computing complexity. We evaluated our architectures on multiple sequence datasets. Of particular note is our performance on the Speech Commands V2 dataset (35 classes). In comparison to SOTA transformer-based models within the ANN domain, our LMUFormer demonstrates comparable performance while necessitating a remarkable $53\times$ reduction in parameters and a substantial $65\times$ decrement in FLOPs. Furthermore, when benchmarked against extant low-complexity SNN variants, our model establishes a new SOTA with an accuracy of 96.12%. Additionally, owing to our model's proficiency in real-time data processing, we are able to achieve a 32.03% reduction in sequence length, all while incurring an inconsequential decline in performance. Our code is publicly available here.

## 1 INTRODUCTION

In recent years, Transformers (Vaswani et al., 2017) have become the most prevalent deep learning models for many applications. The global attention mechanism enables Transformers to capture long-distance dependencies in input data. Moreover, their state-less structure makes their training easily parallelizable, helping them perform well with large-scale datasets, leading to SOTA accuracy in a wide range of applications, including Natural Language Processing (NLP) (Brown et al., 2020; Touvron et al., 2023), Computer Vision (CV) (Dosovitskiy et al., 2020; Liu et al., 2021), and spoken term classification (STC) (Gulati et al., 2020; Gong et al., 2021). However, their self-attention mechanism imposes compute and memory complexity that is quadratic in the sequence length $N$. Moreover, the nature of the self-attention computation prevents the underlying hardware from performing much of the computation until after the entire input sequence is available.

In contrast, Recurrent Neural Network (RNN) models are designed to process data sequentially and their complexity is only $\mathcal{O}(N)$ making them an attractive low-complexity alternative to transformers. However, RNN models traditionally have higher training time as the training procedure must

accommodate the long sequence of dependencies within the model, making parallelization more difficult. Moreover, unlike transformers, RNN models like LSTMs (Hochreiter & Schmidhuber, 1997) and GRUs (Cho et al., 2014) can only leverage past and current information and traditionally suffer from forgetting due to having a limited memory horizon. Recently, however, Legendre Memory Units(Voelker et al., 2019) have been proposed that have the ability to remember information over a theoretically infinite time horizon. Benefiting from this property, LMU-based models outperform many RNN models while still underperforming compared to Transformer alternatives. We hypothesize that the remaining performance gap between LMUs and Transformers is not only because Transformers benefit from future information, but also because they possess more complex network structures, such as the self-attention computation, and thus have higher representational capacity.

In an attempt to explore this hypothesis, we propose a novel sequential network architecture, dubbed LMUFormer, that augments the LMU module with convolutional patch embedding and convolutional channel mixers. Importantly, our convolutional patch embedding only interacts with neighboring input samples and our convolutional channel mixers and the final classifier only operate on the current state of the LMU. Therefore, our LMUFormer model can process data sequentially. Moreover, we present a spiking version of this architecture which extends the benefits of state that is explicit in the LMU implicitly to the patch embedding and channel mixing structures to improve accuracy while simultaneously further enabling complexity reduction. In summary, this paper makes the following contributions:

- We propose a novel architecture LMUFormer which outperforms most existing RNN models with similar complexity on a wide range of sequence learning tasks. On evaluating the Speech Commands dataset, our LMUFormer model, when benchmarked against transformer-based models with competitive performance levels, manifests a significant reduction — accounting for $53\times$ fewer parameters and $65\times$ diminished FLOPs.

- We further devised a spiking variant of the LMUFormer that not only achieves state-of-the-art (SOTA) performance within the realm of SNN models on the Speech Commands dataset but also presents comparable results on the Long Range Arena benchmark.

- Attributable to the real-time data processing capabilities inherent to our models, we evaluate the performance of the model when different proportions of sequences are received. We demonstrate that our model can achieve 99% of the original performance while reducing 32.03% sequence length.

## 2 PRELIMINARIES

**Legendre Memory Unit & Parallel Training:** The Legendre Memory Unit (LMU) is a memory cell proposed by Voelker et al. (2019) designed to efficiently capture and represent temporal dependencies in sequential data rooted in the mathematical properties of Legendre polynomials. Mathematically, the LMU is based on two state-space matrices $(\boldsymbol{A}, \boldsymbol{B})$ that approximate a linear transfer function in continuous time as follows

$$\dot{\boldsymbol{m}}(t) = \boldsymbol{A}\boldsymbol{m}(t) + \boldsymbol{B}\boldsymbol{u}(t), \tag{1}$$

where $u(t)$ represents the input signal and $m(t)$ represents the memory state vector. This continuous-time system is mapped onto discrete time with time step $t$ as follows

$$\boldsymbol{m}[t] = \bar{\boldsymbol{A}}\boldsymbol{m}[t-1] + \bar{\boldsymbol{B}}\boldsymbol{u}[t], \tag{2}$$

where $\bar{\boldsymbol{A}}$ and $\bar{\boldsymbol{B}}$ are the discretized version of $\bar{\boldsymbol{A}}$ and $\bar{\boldsymbol{B}}$.

To better support parallel training, we adopt the same module design as Chilkuri & Eliasmith (2021) to get $u[t]$ from $x[t]$, which is the input signal of the LMU cell in $t$ time step, as follows

$$\boldsymbol{u}[t] = Act_u(\boldsymbol{W}_u\boldsymbol{x}[t] + \boldsymbol{b}_u) \tag{3}$$

and obtain the output of the LMU module as described as follows

$$\boldsymbol{o}[t] = Act_o(\boldsymbol{W}_m\boldsymbol{m}[t] + \boldsymbol{W}_x\boldsymbol{x}[t] + \boldsymbol{b}_o), \tag{4}$$

where $\boldsymbol{W}_u$, $\boldsymbol{W}_m$, and $\boldsymbol{W}_x$ are the learnable parameter matrices. Note that $Act_u$ and $Act_o$ represent the activation functions. Therefore, the module only has one recurrent connection and can be

regarded as a linear time-invariant (LTI) system which can be solved in a non-iterative way which is the key for parallel training. We also adopt fast Fourier transform (FFT) as Chilkuri & Eliasmith (2021) to further reduce the training complexity to $\mathcal{O}(N \log_2(N) \cdot d_c)$, where $N$ is the length of the sequence and $d_c$ is the feature dimension of the input $x$.

**Spiking Neural Network:** As the third-generation neural network (Maass, 1997), SNNs have gained a lot of attention for their potential for higher energy efficiency than traditional ANNs. Mimicking the behavior of biological neurons which communicate using brief electrical pulses, SNNs use binary "spikes" to process and transmit information. However, to convert sequential multi-bit input data such as audio or text into spikes, a coding scheme is required. Rate coding (Lee et al., 2016) and direct coding (Wu et al., 2019) are two of the most commonly used methods. Rate coding translates the input sequence into a spike train across $T$ time steps, with the spike count correlating to input magnitude and spikes following a Poisson distribution (Lee et al., 2020). In direct coding, in contrast, the multi-bit inputs are fed to the first convolution layer in the models and spikes are used only in subsequent portions of the network. If the dataset does not contain temporal information, the outputs of the $1^{st}$ convolution layer need to be repeated $T$ time steps, and converted to binary or multi-bit spikes through spiking neurons.

In this paper, we use direct coding as well as the Leaky Integrate-and-Fire (LIF) (Maass, 1997) neuron model. The behavior of the LIF neuron is described as follows:

$$\boldsymbol{u}_l^t = \lambda \boldsymbol{u}_l^{t-1} + \boldsymbol{w}_l \boldsymbol{o}_{l-1}^t - v_l^{th} \boldsymbol{o}_{l-1}^{t-1} \qquad \boldsymbol{o}_l^{t-1} = \begin{cases} 1, & \text{if } \boldsymbol{u}_l^{t-1} \geq v_l^{th}; \\ 0, & \text{otherwise} \end{cases} \tag{5}$$

$\boldsymbol{u}_l^t$ represents the membrane potential tensor of the $l^{th}$ layer at the $t^{th}$ time step, $\lambda$ is a leak factor that varies between 0 and 1, $\boldsymbol{w}_l$ is the weight connecting layers $(l-1)$ and $l$, $\boldsymbol{o}_{l-1}^t$ is the spike output of the $(l-1)^{th}$ layer at the $t^{th}$ time step, $v_l^{th}$ is the threshold that is kept constant for layer $l$.

## 3 RELATED WORK

**SNN for sequential learning:** In the domain of computer vision, various SNN models have been proposed, serving purposes ranging from image recognition (Fang et al., 2021; Meng et al., 2022) to object detection (Kim et al., 2020; Barchid et al., 2023), and these models have demonstrably achieved competitive performance relative to their ANN counterparts. Notwithstanding, the exploration of SNNs in sequential tasks, such as text classification and spoken term classification, remains notably limited, with scant literature (Lotfi Rezaabad & Vishwanath, 2020; Datta et al., 2023; Lv et al., 2022) addressing these applications.

**Recurrent Transformers:** Since the inception of the work Katharopoulos et al. (2020) that proposed linear transformers, many researchers focused on modifying the self-attention mechanism to make transformer-based model have lower costs during inference. Linformer (Wang et al., 2020) incorporates fixed-size linear projections, facilitating a streamlined approximation of attention across longer sequences. Similarly, Nystromformer (Xiong et al., 2021)leverages the Nyström method to realize an approximation of the standard self-attention mechanism, achieving linear complexity. In a distinct approach, Peng et al. (2023) proposed a novel model architecture, Receptance Weighted Key Value (RWKV). Notably, the time-mixing block within RWKV can arguably be interpreted as computing the product of the matrices K and V, subsequently multiplied by R, suggesting its foundational roots in the Transformer paradigm. Building upon this foundation, Zhu et al. (2023) design the SpikeGPT based on RWKV which has demonstrated competitive performance across both Natural Language Generation (NLG) and Natural Language Understanding (NLU) tasks.

**MLP-Mixer:** The MLP-Mixer (Tolstikhin et al., 2021) has emerged as a novel paradigm in the field of computer vision, diverging from the conventional approaches of Convolutional Neural Networks (CNNs) and Vision Transformers (ViTs). The MLP-Mixer leverages multilayer perceptrons (MLPs) for both spatial and channel-wise data processing. By sequentially employing independent MLPs, it avoids convolution and attention mechanisms, resulting in a streamlined architecture. Inspired by this, we followed a similar framework to design our LMUFormer but further improved it to handle data with temporal information.

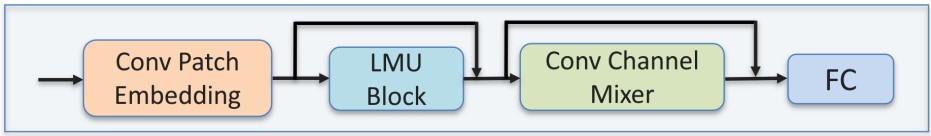

Figure 1: The overall network architectures of LMUFormer.

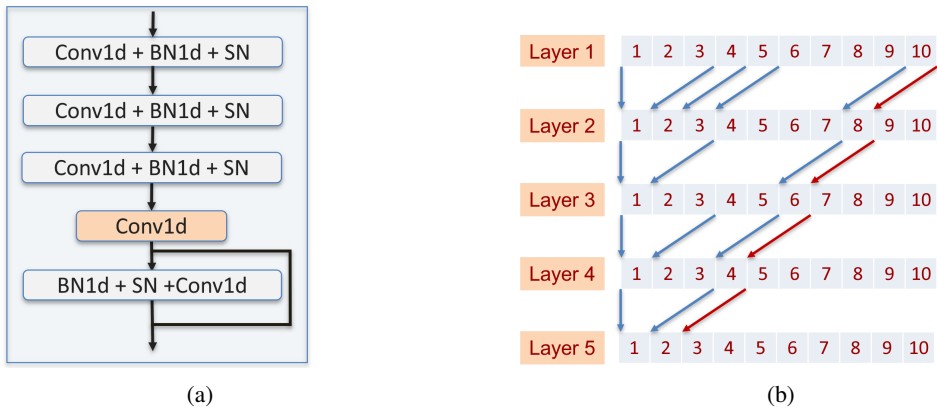

| (a) | (b) |

Figure 2: (a) The structure of the convolutional patch embedding; (b) The delay analysis of the convolutional patch embedding.

# 4 PROPOSED SPIKING LMUFORMER

We propose spiking LMUFormer as shown in Fig. 1, which can be trained in parallel and process data sequentially during inference. Our model is primarily based on LMU and augmented with several techniques from Transformers. Specifically, our model consists of a convolutional patch embedding, LMU block, convolutional channel mixers, and a final fully-connected layer as the classification head. This section first introduces the structure of the patch embedding and channel mixers and then elaborates on how to convert the model to an efficient spiking version.

## 4.1 CONVOLUTIONAL PATCH EMBEDDING

Embedding is the first component of Transformer models and is responsible for converting words/tokens into continuous vectors. For NLP tasks, we simply use the built-in PyTorch embedding module to model the embedding functionality. For STC tasks, we design two types of Convolutional patch embedding. The first is inspired by Gong et al. (2021) and Xiao et al. (2021) which used a series of 2D convolutional layers to split the spectrogram into a sequence of $N$ 16×16 patches. Specifically, we create a new dimension for the embedding features and regard the spectrogram as an image to perform convolution. This structure mixes temporal and frequency information to extract superior feature representations. However, it also yields significant algorithmic delays when processing sequential input data, hindering real-time processing.

To mitigate this concern, we propose to use several 1D convolutional layers with a kernel size of 3 for patch embedding, as we illustrate in Fig. 2(a). When the kernel size is 1, the patch embedding layer is ideal for processing sequential data with no algorithmic latency but also suffers from poor performance. We empirically observed that a convolution kernel of size 3 can dramatically improve the performance since it can capture information about nearby samples and, as analyzed in Fig. 2(b), the added latency is negligible. As an example, the blue lines in Fig. 2(b) represent that to obtain the first output in layer 5, i.e., the output of the patch embedding, we need to wait until the $9^{th}$ input sample arrives. After that, for each input sample fed into the model, the patch embedding can calculate an output, as shown by the red lines. Thus, our network can process the input sequence in real-time after a delay of only 9 samples, which is negligible compared to the total number of samples (typically at least a few hundreds) present in any sequence learning task.

## 4.2 Convolutional Channel Mixer

Inspired by Tolstikhin et al. (2021), we expect the LMU module to be primarily responsible for mixing information across tokens due to the presence of the state variables. To effectively capture the information across the different feature channels, we augment the LMU module with a channel mixer module, that consists of BN and non-linear activation layers followed by $1 \times 1$ convolutional layers as described in Eq. 6. The first activation layer employs Gaussian Error Linear Units (GELU) as introduced by Hendrycks & Gimpel (2016), while the subsequent activation layer utilizes the Rectified Linear Units (ReLU) (Krizhevsky et al., 2012). Since the convolutional layer, BN layer, and non-linear activation layer do not interact with temporal information, we omit the time notation $T$ in Eq. 6 for simplicity. We also add a residual connection (He et al., 2016) between the input $X_i$ and the output $X_o$ of the convolutional channel mixer and refer to this enhanced structure as the Conv Channel Mixer Block.

$$X = Conv_1(GELU(BN_1(X_i))), \quad X_o = Conv_2(ReLU(BN_2(X))) \tag{6}$$

## 4.3 Spiking LMUFormer Network Structure

To further improve the energy efficiency, we propose a spiking version of the LMUFormer, named spiking LMUFormer.

For NLP tasks, we use the batch normalization and the spiking LIF (SN) layer in the first LMU block to convert the floating point values to spikes, that results in sparsity and accumulate-only operations. For STC tasks, we use the first convolution layer together with its followed BN and SN layer in the patch embedding as the direct encoder. Inspired by (Yao et al., 2023), we also adjust the positions of residual shortcuts to make sure the output is the addition between two floating point numbers rather than the spikes.

As depicted in Fig. 3, the spiking LMU block encompasses several layers, structured sequentially as: a BN layer, followed by a SN layer, succeeded by the core spiking LMU cell, which is then coupled with a Conv layer, and concluded with another BN layer. Notably, the integration strategy we employed capitalizes on the inherent temporal dynamics of both the LMU and the SNN. Considering the concurrent updates of both the memory and hidden states in the LMU at every discrete time step, in conjunction with the analogous updates of the membrane potentials and spikes within the SNN, we have devised a merged process to optimize the overall operational efficiency of their integration.

In particular, we feed the input spikes $X_S[t]$ into a convolutional layer and a BN layer to get the input signal of the memory cell.

$$U[t] = BN(Conv1d(X_S[t])) \tag{7}$$

Meanwhile, they are also regarded as the input of spike neurons. As we adopt the Leaky Integrate-and-Fire (LIF) neuron (Maass, 1997) model, the update of the membrane potential $U_H[t]$ at time step $t$ is described below (Wu et al., 2018)

$$U_H[t] = U_V[t-1] + \frac{1}{\tau}(U[t] - (U_V[t-1] - U_V^{reset})), \tag{8}$$

where $U_V[t-1]$ means the membrane potential after emitting spikes at time step $t-1$, $\tau$ is the time constant, and $U_V^{reset}$ represents the reset potential. The firing of spikes is then described as follows

$$U_S[t] = F(U_H[t] - U_V^{th}) \tag{9}$$

where $U_S[t]$ means the spikes of the input signal of the memory cell and $F$ denotes the firing function, which outputs 1 when the input is greater than 0, and 0 otherwise. Finally, to reset the membrane potential $U_V$ at time step $t$ we use the following equation

$$U_V[t] = U_H(t) \cdot (1 - U_S[t]) + U_V^{reset} \cdot U_S[t]. \tag{10}$$

After obtaining $U_S[t]$ and the memory vector $M[t-1]$ in the last time step, we can formulate the update of $M[t]$ as shown below

$$M[t] = \bar{A} \cdot M_S[t-1] + \bar{B} \cdot U_S[t], \tag{11}$$

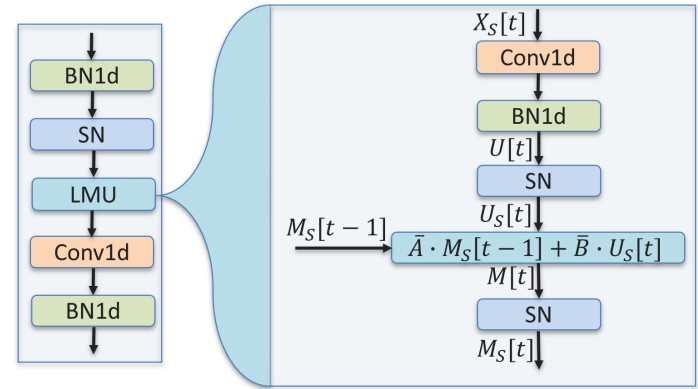

Figure 3: The network architectures of spiking LMU block and the details within the spiking LMU cell.

where $\bar{A}$ and $\bar{B}$ are the discretized space metric from Voelker et al. (2019). To simplify the notation, we use the function Spiking Neuron $SN(\cdot)$ as the abbreviation of Eqs. (8) to (10). Therefore, as shown below

$$M_S[t] = SN(M[t]), \quad O[t] = SN(BN(Conv1d([M_S[t], X_S[t]]))) \tag{12}$$

we first feed the $M[t]$ into the spiking neurons to get the spikes of the memory state at time step $t$. Then we concatenate $M_S[t]$ and $X_S[t]$ along the feature dimension and feed them into the 1D Conv layer, BN layer, and SN to get the output spikes of the LMU block at time step $t$.

## 5 EXPERIMENTS

### 5.1 PERMUTED SEQUENTIAL MNIST

To evaluate the performance of our models on vision task, we use the permuted sequential MNIST (psM-NIST) dataset (Le et al., 2015). Unlike the original MNIST dataset (Le-Cun et al., 1998) showing each $28 \times 28$ grayscale image as a whole, psM-NIST presents the pixels of each image as a sequence, typically in a permuted order. It results in this task being much more challenging than the original task since the models need to remember and integrate in-

| Model | SNN | Acc (%) |
|-------|-----|---------|
| LSTM (Voelker et al., 2019) | N | 89.86 |
| LMU (Voelker et al., 2019) | N | 97.15 |
| HiPPO-LegS (Gu et al., 2020) | N | 98.3 |
| PLMU (Chilkuri & Eliasmith, 2021) | N | 98.49 |
| LMUFormer | N | **98.55** |
| LMUFormer | Y | 97.92 |

Table 1: Performance comparison of our LMUFormer models and previous works on psMNIST dataset.

formation over time, i.e., have the ability to handle long-range dependencies. psMNIST contains 10 classes, involving handwritten digits from 0 to 9 and we split 50k images as training dataset, 10k images as validation dataset, and the rest 10k images as test images.

As shown in Table .3, our models not only outperform all the RNN models but also surpass the existing LMU-based models. Since the psMNIST dataset is relatively small, directly applying our LMUFormer model could easily lead to overfitting, so we used a simplified version. We only use one Conv layer as well as a BN layer as the patch embedding and remove the channel mixer.

Table 2: Performance comparison of our LMUFormer models and previous works on Speech Commands dataset. (Spikformer* and Spike-driven ViT* represent models in which we replace our LMU module with Spiking Self Attention Zhou et al. (2022) and Spike-Driven Self-Attention Yao et al. (2023) modules, respectively.)

| Model | Sequential Inference | Parallel Training | SNN | Accuracy (%) |
|---|---|---|---|---|
| RNN (Bittar & Garner, 2022) | Yes | No | No | 92.09 |
| Attention RNN (De Andrade et al., 2018) | No | No | No | 93.9 |
| liBRU (Bittar & Garner, 2022) | Yes | No | No | 95.06 |
| Res15 (Vygon & Mikhaylovskiy, 2021) | Yes | Yes | No | 97.00 |
| KWT2 (Berg et al., 2021) | No | Yes | No | 97.74 |
| AST (Gong et al., 2021) | No | Yes | No | 98.11 |
| LIF (Bittar & Garner, 2022) | Yes | Yes | Yes | 83.03 |
| SFA (Salaj et al., 2021) | Yes | No | Yes | 91.21 |
| Spikformer* (Zhou et al., 2022) | No | Yes | Yes | 93.38 |
| RadLIF (Bittar & Garner, 2022) | Yes | No | Yes | 94.51 |
| Spike-driven ViT* (Yao et al., 2023) | No | Yes | Yes | 94.85 |
| LMUFormer | Yes | Yes | No | **96.53** |
| LMUFormer (with states) | Yes | Yes | No | **96.92** |
| Spiking LMUFormer | Yes | Yes | Yes | **96.12** |

## 5.2 SPEECH RECOGNITION

We use the Speech Commands V2 dataset (Warden, 2018) contains 105,829 audio files of 35 different words. Each recording lasts up to 1 second and the sampling rate is 16KHz. We use the official dataset splits, where the training, validation, and test dataset contains 84,843, 9,981, and 11,005 samples, respectively. We chose the hardest classification task with 35 classes to test our models.

As shown in Table 2, our LMUFormer model outperforms the existing RNN models. Although there is still about a 1.5% drop in test accuracy between our model with the SOTA transformer-based model, our model has fewer parameters (1.622 Million vs. 86.933 Million, $86.933/1.622 \approx 53.66$), fewer FLOPs ($1.89 \times 10^8$ vs. $1.24 \times 10^{10}$, $124/1.89 \approx 65.61$) and can handle data sequentially. Additionally, our evaluations indicate that the LMUFormer (with states) achieves a noteworthy performance of 96.92%. This empirical outcome substantiates our initial hypothesis that the integration of states within patch embedding and the channel mixer block can enhance model performance. In pursuit of unparalleled energy efficiency, we convert this model to the 1-bit SNN model, i.e., the spiking LMUFormer, which surpasses all the contemporary SNN models with mere $3.09 \times 10^7$ theoretical synaptic operations (SOPs) (Merolla et al., 2014).

In addition, we also added the result of two experiments for two transformer-based SNN models, Spikeformer (Zhou et al., 2022) and Spike-driven ViT (Yao et al., 2023), that achieve the SOTA performance in the vision tasks. Because they do not provide the official version of their models for the SC dataset, we tested new models in which we use their proposed structure of self-attention as a substitute for the LMU module in our model. It is important to note that the

| Model | Params. (M) | OPs (G) |
|---|---|---|
| AST (Gong et al., 2021) | 86.93 | 12.4 |
| LMUFormer | 1.62 | 0.189 |
| Spiking LMUFormer | 1.69 | 0.0309 |

Table 3: Comparison of the number of model parameters and operational counts between our LMUFormer models and AST model. (OPs refers to FLOPs in ANNs, and SOPs in SNNs)

Spikformer (Zhou et al., 2022) model has a dramatic drop compared with the non-spiking transformer-based AST (Gong et al., 2021). This suggests that if we restrict the patch embedding to only mix the information from neighboring samples, the power of the original global attention in transformer models will be degraded. On the contrary, the Spike-driven ViT (Yao et al., 2023) can preserve a higher accuracy which indicates its linear attention is more robust to the degradation of the features extracted by the patch embedding. Moreover, our spiking LMUFormer achieves

Table 4: Performance comparison of our LMUFormer models and previous works on Long Range Arena (LRA) Dataset.

| Model | ListOps (2K) | Text(4K) | Retrieval (4K) | Image(1K) | Pathfinder (1K) | Avg |
|---|---|---|---|---|---|---|
| S4 | 58.35 | 76.02 | 87.09 | 87.26 | 86.05 | 80.48 |
| Linear Trans. | 16.13 | 65.90 | 53.09 | 42.34 | 75.30 | 50.55 |
| Linformer | 35.70 | 53.94 | 52.27 | 38.56 | **76.34** | 51.36 |
| Transformer | 36.37 | 64.27 | 57.46 | 42.44 | 71.40 | 54.39 |
| BigBird | 36.05 | 64.02 | 59.29 | 40.83 | 74.87 | 55.01 |
| Nystromformer | 37.15 | 65.52 | 79.56 | 41.58 | 70.94 | 58.95 |
| LMUFormer | 34.43 | **68.27** | 78.65 | 54.16 | 69.9 | 61.08 |
| Spiking LMUFormer | **37.30** | 65.80 | **79.76** | **55.65** | 72.68 | **62.24** |

Table 5: Ablation study on different patch embeddings and channel mixers in the LMUFormer model in Speech Commands V2 dataset. ( "Naive LMU" means only using the LMU cell as the token mixer while "LMU Block" means adding a Conv layer and BN layer after the LMU cell. "Conv PE" means our convolutional patch embedding and "Conv CM" means our convolutional channel mixer.)

| Model type | Patch Embedding | Token Mixer | Channel Mixer | Acc. (%) | $\Delta$ (%) |
|---|---|---|---|---|---|
| Non-spiking | Identity | Naive LMU | Identity | 76.17 | 0.0 |
| Non-spiking | One-layer Conv | Naive LMU | Identity | 89.28 | 13.11 |
| Non-spiking | One-layer Conv | LMU Block | Identity | 89.8 | 0.52 |
| Non-spiking | Conv PE | LMU Block | Identity | 95.57 | 5.77 |
| Non-spiking | Conv PE | LMU Block | Conv CM | **96.53** | 0.96 |
| Spiking | One-layer Conv | Naive LMU | Identity | 83.78 | 0.0 |
| Spiking | One-layer Conv | LMU Block | Identity | 87.88 | 4.1 |
| Spiking | Conv PE | LMU Block | Identity | 95.20 | **7.32** |
| Spiking | Conv PE | LMU Block | Conv CM | **96.12** | 0.92 |

a further 1.27% increase in accuracy, demonstrating that our model is superior to the traditional transformer-based spiking models when handling information with temporal information.

## 5.3 LONG RANGE ARENA BENCHMARK

To showcase the capability of our LMUFormer with longer tokens, we utilize the Long Range Arena (LRA) benchmark introduced by Tay et al. (2020). The LRA benchmark assesses machine learning models on long-context understanding through subtasks including text classification, document retrieval, image classification, pathfinder, and listops, highlighting their performance across various domains. Adhering to the evaluation protocol from Tay et al. (2020), which establishes specific train/test splits, we report the classification accuracy for each task and present an aggregated performance measure across all tasks.

We conducted a comparative study involving five transformer-based models: the vanilla transformer (Vaswani et al., 2017), Linear Trans. (Katharopoulos et al., 2020), Linformer (Wang et al., 2020), BigBird (Zaheer et al., 2020) and Nystromformer (Xiong et al., 2021). And the results of the first four models are from Tay et al. (2020). As detailed in Table 4, there is still a significant gap in the performance between our models with the recent S4-based models (Gu et al., 2021), but our LMUFormer excels against these transformer-based models in almost all tasks except Pathfinder. Furthermore, our spiking variant of LMUFormer surpasses them in the ListOps and Retrieval tasks.

Notably, our spiking LMUFormer is the inaugural SNN model to not only demonstrate comparable performance on the LRA dataset but also to outshine the majority of its transformer-based counterparts. Intriguingly, the spiking LMUFormer outperforms the regular LMUFormer by an average margin of 0.53%, suggesting the potential of SNN models to harness their inherent temporal dynamics for superior performance with long sequences.

## 5.4 ABLATION STUDY

We first conducted extensive ablation studies to show the impact of the different patch embeddings and channel mixers in our proposed LMUFormer model on the final performance. All the models have one final linear layer that acts as the classification head and are trained and tested on the Speech Command V2 dataset with 35 classes. As shown in Table 5, a naive LMU module with a classification head, same as a pLMU, can only achieve 76.17% accuracy while simply adding a convolution layer can boost the result to 89.28%. Although the performance of the non-spiking model with our LMU block is only slightly better than the model with a naive LMU, our LMU block significantly improves the performance of the spiking model by 4.1% (an increase from 83.78% to 87.88%). Moreover, the results show our convolutional patch embedding can further improve the accuracy of the models compared to models with a simple convolution layer. Finally, using a convolutional channel mixer, our models achieve a final accuracy of 96.53% for the non-spiking model and 96.12% for the spiking model.

## 5.5 RECURRENT PROCESSING

To more clearly showcase our model's proficiency with sequential data, we evaluated the trained spiking LMUFormer on the Speech Command V2 test dataset, gradually increasing the sequence length from 0 to its full length of 128 samples. As shown in Fig. 4, the increase in the model's accuracy with the number of samples first accelerates continuously and then levels off. This indicates that after a certain number of samples have been obtained, the model has been able to predict the results almost correctly. Specifically, our model achieves 99% of its original performance while getting a 32.03% (1 - 87/128) reduction in the sequence length, yielding results of 95.17% compared to 96.12% which is even higher than the spikformer model (93.38%) which utilizes all the information from the whole sequence.

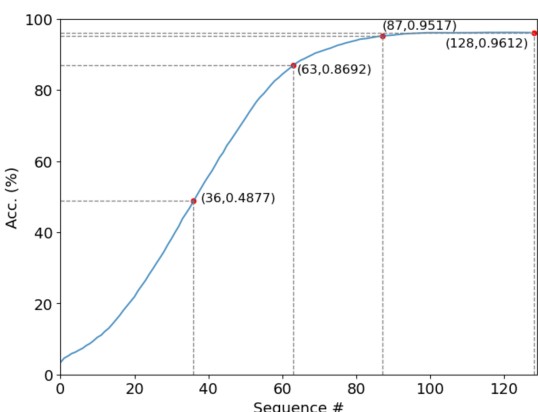

Figure 4: Plot of test accuracy as the number of samples in the sequence increases.

## 6 CONCLUSIONS

The transformer model excels across various domains, particularly with large-scale datasets, owing to its exceptional performance and straightforward parallel training. However, its $\mathcal{O}(N^2)$ complexity during inference, coupled with the necessity to acquire the complete sequence before calculating the self-attention, limits its utility in resource-constrained environments demanding low latency. Hence, we introduce the LMUFormer and Spiking LMUFormer models, uniquely designed to switch between parallel and recurrent forwarding when training and testing. Through extensive experiments, we demonstrate that our non-spiking model achieves close to the performance of state-of-the-art (SOTA) models while markedly reducing the model size by approximately $53\times$ and computational complexity by roughly $65\times$. Furthermore, our Spiking LMUFormer achieves SOTA performance, registering a notable accuracy of 96.12% among prevailing SNN models evaluated on the Speech Commands V2 dataset, without compromising on efficiency. It is our aspiration that this contribution serves as a catalyst for further exploration and advancements in model design for SNNs, particularly in the domain of sequence learning tasks.

## 7 ACKNOWLEDGEMENTS

This work is supported by a gift funding from Intel Labs. We would also like to thank Sumit Bam Shrestha and Timothy Shea from Intel for insightful discussions on LMUFormer.

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

Table 6: The number of batch size and maximum learning rate for different subtasks in the LRA dataset. The number in parentheses indicates the learning rate for spiking models if it is different from the learning rate of the non-spiking models.

| Hyperparameters | ListOps (2K) | Text(4K) | Retrieval (4K) | Image(1K) | Pathfinder (1K) |
|---|---|---|---|---|---|
| Batch size | 32 | 32 | 32 | 256 | 256 |
| Max. Learning rate | 0.01 | 0.0001 | 0.0001(0.001) | 0.01 | 0.001 |

Table 7: Performance comparison of our LMUFormer models and previous works on the LRA dataset.

| Model | ListOps (2K) | Text(4K) | Retrieval (4K) | Image(1K) | Pathfinder (1K) | Avg |
|---|---|---|---|---|---|---|
| LMUFormer | 34.43 | **68.27** | 78.65 | 54.16 | 69.9 | 61.08 |
| Spiking LMUFormer | **37.30** | 65.80 | **79.76** | **55.65** | **72.68** | **62.24** |
| pLMU | 22.83 | 63.72 | 74.1 | 53.50 | 68.32 | 56.49 |
| Spiking pLMU | 17.94 | 60.84 | 65.19 | 51.89 | 49.88 | 49.15 |

# A  APPENDIX

## A.1  LRA BENCHMARK

The benchmark comprises:

- **ListOps**: A synthetic task where models operate on nested lists, gauging their understanding of hierarchical structures.
- **Text**: Byte-level classification of IMDb reviews, pushing the boundaries on sequence lengths.
- **Retrieval**: Document retrieval tasks centered around byte-level tokens.
- **Image**: Image classification, but uniquely handled as pixel sequences, sidestepping traditional 2D processing.
- **Pathfinder**: An image-based task assessing if an unobstructed path connects two highlighted points, challenging models on long-range spatial dependencies.

## A.2  TRAINING HYPERPARAMETERS

For psMNIST dataset, we trained the models 50 epochs with Adam (Kingma & Ba, 2014) optimizer and the initial learning rate is 0.0001 and the batch size is 100. As for the Speech Commands V2 dataset, the batch size is set to 128 during 100 (200) epochs with an initial learning rate of 0.00025 (0.0025) and weight decay of $5e-7$. After the very first 5 (10) epochs, we multiply the learning rate by 0.85 every 5 (10) epochs. The numbers in brackets are the settings for spiking models. For the LRA dataset, we generally used the setting as Gong et al. (2021). For all subtasks, we adopted AdamW (Loshchilov & Hutter, 2017) optimizer and the 1cycle learning rate policy (Smith & Topin, 2019). The weight decay for the 'Image' task is 0.01 and 0 for other tasks. More hyperparameters are shown in Table. 6.

## A.3  MODEL BIAS & LIMITATIONS

There may be a few potential limitations or scenarios where LMUFormer may not perform very well. For example, LMUFormer may lack the pre-training capabilities of a Transformer, that can enable high performance when fine-tuned on a range of downstream tasks. Rather, it needs to be trained on the streaming use-case directly by the user. That said, it is not very clear whether pre-training on a large text corpus that empower Transformers, can improve the performance of our models on streaming use-cases. Moreover, to our best knowledge, there is no large-scale streaming dataset, such as the BookCorpus, that we can use to pre-train our models. Given a large-scale pre-trained

Table 8: Performance, training time and memory comparison on LRA 'Text' subtask.

| Model | Text(4K) | Params. (M) | Peak memory (MB) | Training (s) | Validation (s) |
|---|---|---|---|---|---|
| Linear Trans. | 65.90 | 0.174 | 644 | 5.91 | 2.37 |
| Linformer | 53.94 | 0.698 | 1180 | 8.69 | 2.46 |
| Transformer | 64.27 | 0.174 | 1780 | 20.60 | 2.49 |
| Nystromformer | 65.52 | 0.174 | 836 | 11.24 | 4.09 |
| LMUFormer | 68.27 | 0.174 | 738 | 6.29 | 2.46 |
| Spiking LMUFormer | 65.80 | 0.175 | 928 | 12.02 | 5.66 |
| pLMU | 63.72 | 0.132 | 582 | 4.60 | 1.83 |
| Spiking pLMU | 60.84 | 0.132 | 684 | 8.68 | 3.75 |

dataset (and enough compute power), it may be possible for LMUFormer to scale up, and achieve high performance when fine-tuned on downstream tasks. We hypothesize this may not be possible by simply stacking LMUFormer blocks, and may require network architectural optimizations, which is an important research direction.

Moreover, as shown in Table 4, LMUFormer, though yields higher performance compared to all transformer variants, can not surpass the S4 models on LRA dataset, that evaluates model quality under long-context scenarios. Improving the LMUFormer performance for such datasets is also an important and interesting research direction.

## A.4 COMPARISON BETWEEN DIFFERENT MODELS ON LRA

For the LRA dataset, we use the same word and position embedding for all the models, replacing the patch embedding block we use for LMUFormer. This narrows the gap between LMUFormer and pLMU models on LRA compared to Google Speech Commands as shown in Table 7. Overall, the LMUFormer significantly outperforms pLMU, with comparable results only in Task 'Image' and Task 'Pathfinder'.

We also show a comparison of performance, model size, peak memory, training time, and inference time for the pLMU model, LMUFormer, Spiking LMUFormer and other Transformer variants, using the "Text" subtask as an example. Our results are shown in the Table 8. As we can see, our LMU-Former achieves the best performance while maintaining relatively decent training and inference speeds, and requires less memory during training.

## A.5 HARDWARE-SOFTWARE CO-DESIGN

We develop a hardware simulation framework for SNNs to estimate the energy, latency, and throughput of our non-spiking and spiking LMUFormer models. We have also incorporated the overhead due to the spike sparsity in our framework, which is minimal with high sparsity as obtained in this work.

The total compute energy (CE) of the spiking LMUFormer ($SpLMU_{CE}$) can be estimated as

$$SpLMU_{CE} = \sum_{t=1}^{T} \left( DNN_1^{op} E_{mac} + \sum_{l=2}^{L} (S_l^t DNN_l^{op} E_{ac} + E_{sp} DNN_l^{op}) + \sum_{l=1}^{L} DNN_l^{com} E_{com} \right)$$
(13)

because the SNN receives full-precision input in the first layer ($l=1$) without any sparsity. Note that $DNN_l^{com}$ denotes the total number of comparison (thresholding) operations in the layer $l$ with each operation consuming 1.64pJ energy in our 28nm Kintex-7 FPGA platform for floating point (FP) reperesentation. Also, note that $DNN_l^{op}$ denote the total number of floating point (MAC or AC) operations in layer $l$. Lastly, $S_l^t$ denotes the activation sparsity of layer $l$ at time step $t$, and $E_{sp} = 0.05$pJ denotes the energy overhead due to sparsity, that is incurred in checking whether the binary activation is zero.

The CE of the full-precision LMUFormer ($LMU_{CE}$) is estimated as $DNN_{CE} = \sum_{l=1}^{L} DNN_{l}^{op} E_{mac}$, where we ignore the energy consumed by the non-linear activation operation (significantly lower compared to thresholding operation).

The compute-efficiency of the spiking LMUFormer stems from two factors: 1) high activation sparsity, where $\sum_{t=1}^{T} S_{l}^{t}$=0.15 on average across all the layers, and 2) Use of only AC (1.8pJ) operations that consume $7.4\times$ lower compared to each MAC (13.32pJ) operation in our FPGA setup for floating point (FP) representation. Note that the binary activations can replace the FP multiplications with logical operations, i.e., conditional assignment to 0 with a bank of AND gates. These replacements may be realized using existing hardware depending on the compiler and the details of their data paths. Based on these two factors, we observe that our spiking LMUFormer incurs $27.2\times$ lower compute energy compared to LMUFormer on the Google speech commands at iso-parameter.

In contrast, the energy incurred in the memory access of the weights for both non-spiking and spiking LMUFormer depend on the data re-use scheme and the underlying hardware. However, since both the models have almost the same number of trainable parameters, they are expected to incur identical memory energy. We also do not expect any additional latency or throughput improvement in SNN, since we need to process the identical sequential input, and activation sparsity favors compute energy (and not latency). That said, unlike existing SNNs that incur an additional temporal overhead and suffers from high latency, our SNN re-uses the hidden memory dimension of the LMUformer to incur the temporal dimension. Thus, our spiking LMUFormer can yield higher compute efficiency with no overhead on latency.

