# OpenReview forum: "LMUFormer: Low Complexity Yet Powerful Spiking Model With Legendre Memory Units"
_ICLR.cc/2024/Conference — ICLR 2024 poster_

### Official Review · Reviewer_4AC7 · 2023-10-31

**Soundness:** 3 good
**Presentation:** 3 good
**Contribution:** 3 good
**Rating:** 6
**Confidence:** 2

**Summary:**

Transformer models are highly accurate but complex and not suited for sequential processing, making them ill-suited for edge devices. RNNs are more suitable for these devices due to their lower complexity and sequential processing capabilities, but they often lag in performance compared to Transformer models. The paper proposes LMUFormer, a model that augments Legendre Memory Units (LMU) with convolutional patch embedding and convolutional channel mixers. This results in a fully-sequential recurrent model that approaches the performance of Transformer models while retaining the ability to process data sequentially.  A spiking version of LMUFormer is also introduced, which brings the benefits of states within the patch embedding and channel mixer modules, further reducing computing complexity.
The contribution lies in
1. New Model Architecture: Introduction of LMUFormer, a novel architecture that combines LMUs with convolutional patch embedding and channel mixers, aiming to provide a balance between high performance and low complexity.

2. Performance: LMUFormer demonstrates impressive performance, particularly on the Speech Commands V2 dataset, showing comparable results to state-of-the-art transformer-based models but with significantly fewer parameters and lower computational complexity.

3. Spiking Version: Presentation of a spiking version of LMUFormer, establishing a new state-of-the-art in low-complexity Spiking Neural Network (SNN) variants with an accuracy of 96.12%, and demonstrating the ability to process real-time data efficiently.

4. Efficiency in Real-Time Processing: The model shows proficiency in real-time data processing, achieving a 32.03% reduction in sequence length with minimal performance decline, highlighting its suitability for streaming applications at the edge.

**Strengths:**

**Originality:**
- Innovative Model Design: The LMUFormer introduces a unique combination of Legendre Memory Units (LMU) with convolutional patch embedding and channel mixers, creating a novel architecture that stands out in the realm of spiking models and RNNs.
- Spiking Model Integration: The integration of a spiking version of the LMUFormer adds a layer of originality, as it brings the benefits of states within the patch embedding and channel mixer modules, while also aiming to reduce computing complexity.

**Quality:**
- Robust Performance: The LMUFormer demonstrates robust performance, especially highlighted by its results on the Speech Commands V2 dataset, where it shows comparable performance to state-of-the-art transformer-based models but with a significant reduction in parameters and computational complexity.
- Comprehensive Evaluation: The paper includes a thorough evaluation of the architectures on multiple sequence datasets, providing a solid basis for the claims made about the model’s performance and efficiency.

**Clarity:**
- Well-Structured: The paper is well-structured, with clear sections that logically flow from one to the next, making it easy for readers to follow the development of ideas and understand the proposed model.
- Detailed Explanations: The authors provide detailed explanations of the LMUFormer architecture, the spiking version of the model, and the motivations behind their design choices, contributing to the overall clarity of the paper.

**Significance:**
- Addressing Resource Constraints: The LMUFormer addresses a significant challenge in the field of edge computing and streaming applications, where devices are heavily resource-constrained. By providing a model that combines high performance with low complexity and sequential processing capabilities, the paper makes a meaningful contribution to this area.

**Weaknesses:**

**Addressing Potential Biases:**
Model Limitations: The paper could be improved by providing a more balanced view, including a discussion of potential limitations or scenarios where the LMUFormer might not perform as well. This would help readers develop a more nuanced understanding of the model’s applicability.

**Enhancing Reproducibility:**
Implementation Details: Providing more implementation details, including hyperparameters and training procedures, would enhance the reproducibility of the results, contributing to the paper’s overall quality.

**Real-Time Processing Analysis:**
Given the focus on streaming applications and real-time data processing, a more detailed analysis of the model’s performance in real-time scenarios, including potential latency issues and how they are addressed, would be valuable.

**Software-Hardware codesign**
It would be nice to see a hardware simulation for SNN to learn all aspect of the new model, engergy, latency, throughput and any overhead, etc.

**Questions:**

- citation for S4 models?

- Could you add "how to get 32.03%" in section 5.5? I assume it is (128-87)/128? similarly to 70x and 140x to help reading

---

> ### Author Response · Authors · 2023-11-17
> **Response to Reviewer 4AC7**
>
> Thanks for your positive comments and valuable suggestions to improve the quality of our work. Please see our response below.
>
> **Addressing Potential Biases**
>
> Thank you for bringing up this important point. There may be a few potential limitations or scenarios where LMUFormer may not perform very well. For example, as mentioned by Reviewer **KEVY**, LMUFormer may lack the pre-training capabilities of a Transformer, that can enable high performance when fine-tuned on a range of downstream tasks. Rather, it needs to be trained on the streaming use-case directly by the user. Developing architectural optimizations for LMUFormer to scale up and benefit from pre-training is an important research direction.
>
> Moreover, as shown in Table 4, LMUFormer, though yields higher performance compared to all transformer variants, can not surpass the S4 models on LRA dataset, that evaluates model quality under long-context scenarios. Improving the LMUFormer performance for such datasets is also an important and interesting research direction.
>
> These limitations have been added in Appendix A.2 of the revision.
>
> **Implementation Details**
>
> Thank you for the important reminder. We have added the training configuration for the three tasks considered in this work in appendix A.1.
>
> **Real-Time Processing Analysis**
>
> So far, we have only explored the real-time processing theoretically; we show that our model can process each sample as soon as it arrives (after accumulating a tiny fraction of the total number of samples), which is not feasible with the traditional Transformer models. A deeper analysis may necessitate considering a variety of practical settings that may impact the real-time processing capability. For example, the hardware processing the model need to be energy efficient yet provide high utilization to ensure each input sample is accepted promptly by the model once it arrives. Moreover, the latency of the model would depend on the communication overhead of the sensor generating the input sample. To minimize the latency and enable real-time processing, this overhead should not exceed the time required by the model to process of one sample. The accurate real-time processing analysis will thus depend on the both the underlying sensor and deep learning hardware as well as the dataflow.
>
> **Hardware-Software Co-Design**
>
> Thanks for your suggestion. We have now developed a hardware simulation framework for SNNs to analyze the energy, latency, and throughput of our non-spiking and spiking LMUFormer models. We have also incorporated the overhead due to the spike sparsity in our framework, which is minimal with high sparsity as obtained in this work. Our SNNs yield $27.2\times$ lower compute energy compared to LMUFormer on the Google speech commands with no additional overhead on latency and throughput, since our SNNs do not incur any additional overhead, unlike traditional SNNs.
>
>
> **Citation for S4 models, and "how to get 32.03%" in section 5.5? Is it (128-87)/128? How 70x and 140x**
>
> We really appreciate you catching the two errors in the paper; we have now added the citation for S4 models and the details of how we arrived at 32.03% reduction in sequence length. You are right that it is indeed (128-87)/128.
>
> We also apologize for a mistake we had made in the paper regarding the model size and FLOPs reduction factors. We actually showed the correct absolute model size and FLOPs numbers for the two different models, but made a mistake while computing the reduction factors. The correct reduction factor of our LMUFormer compared to the SOTA AST model is $86.933/1.622\approx53.66$ for the model size and $124/1.89\approx65.61$ for FLOPs. We have corrected these numbers in the revision.

---

### Official Review · Reviewer_FTnd · 2023-11-01

**Soundness:** 2 fair
**Presentation:** 3 good
**Contribution:** 2 fair
**Rating:** 6
**Confidence:** 2

**Summary:**

This paper proposes a LMU based model that also has a spiking variant. The goal is to explore the accuracy gap between transformer based model with LMU based model with more complex sub-modules.

**Strengths:**

* This paper is mostly easy to follow and well organized.
* Related background and works are discussed and introduced extensively.
* The intention of exploring this performance gap in function complexity feels intuitive and interesting.
* Experimental are conducted extensively amongst different models and tasks and comparing both accuracy and computational cost, results also look mostly promising.

**Weaknesses:**

* In task 2, there is still some accuracy gap between the proposed method and some transformer based methods. Is there any way to compare pLMU on the same task as well as resource usage? It feels the narrative is that accuracy wise on relative complex tasks LMUformer is on par with transformer based method while out-performing pLMU. Besides rMNIST task, it is difficult to tell how much performance gain that the proposed method would offer while (potentially) introducing extra hardware cost.

**Questions:**

* Does the proposed method suit SNN better and more nature? What about the training cost and time?

---

> ### Author Response · Authors · 2023-11-17
> **Response to Reviewer FTnd**
>
> Thanks for your positive comments and valuable suggestions to improve the quality of our work. Please see our response below.
>
> **Comparison between LMUFormer and pLMU**
>
> We would like to clarify that we compare the performance of pLMU and our LMUFormer in Table 1 for psMNIST and Table 5 for Speech Commands. Note that in Table 5, the model in the first row actually represent the pLMU model, and we have clarified them in the revision. In the revision, we have added their comparison for the LRA dataset, which is shown in the table below.
>
> | Model              | ListOps (2K) | Text (4K) | Retrieval (4K) | Image (1K) | Pathfinder (1K) | Avg   |
> |--------------------|--------------|-----------|----------------|------------|-----------------|-------|
> | LMUFormer          | 34.43        | **68.27** | 78.65          | 54.16      | 69.9            | 61.08 |
> | Spiking LMUFormer  | **37.30**    | 65.80     | **79.76**      | **55.65**  | **72.68**       | **62.24** |
> | pLMU               | 22.83        | 63.72     | 74.1           | 53.50      | 68.32           | 56.49 |
> | Spiking pLMU       | 17.94        | 60.84     | 65.19          | 51.89      | 49.88           | 49.15 |
>
>  For the LRA dataset, we use the same word and position embedding for all the models, replacing the patch embedding block we use for LMUFormer. This narrows the gap between LMUFormer and pLMU models on LRA compared to Google Speech Commands. Overall, the LMUFormer significantly outperforms pLMU, with comparable results only in Task 'Image' and Task 'Pathfinder'. Note the results on 'Image' task are different from the initial version of the paper, since we changed the initial learning rate and weight decay to improve the performance.
>
> We also show a comparison of performance, model size, peak memory, training time, and inference time for the pLMU model, LMUFormer, Spiking LMUFormer and other Transformer variants, using the "Text" subtask as an example. Our results are shown in the Table below:
>
>  | Model             | Text(4K) | Params. (M) | Peak memory (MB) | Training time (s) | Validation time (s) |
> |-------------------|----------|-------------|------------------|--------------------|---------------------|
> | Linear Trans.     | 65.90    | 0.174       | 644              | 5.91               | 2.37                |
> | Linformer         | 53.94    | 0.698       | 1180             | 8.69               | 2.46                |
> | Nystromformer     | 65.52    | 0.174       | 836              | 11.24              | 4.09                |
> | LMUFormer         | 68.27    | 0.174       | 738              | 6.29               | 2.46                |
> | Spiking LMUFormer | 65.80    | 0.175       | 928              | 12.02              | 5.66                |
> | pLMU              | 63.72    | 0.132       | 582              | 4.60               | 1.83                |
> | Spiking pLMU      | 60.84    | 0.132       | 684              | 8.68               | 3.75                |
>
> As we can see, our LMUFormer achieves the best performance while maintaining relatively decent training and inference speeds, and requires less memory during training.
>
> **Suitability of LMUFormer with SNNs**
>
> LMUFormer has recurrent processing capabilities similar to SNNs. However, naively applying the SNN to LMUFormer would incur an additional temporal dimension where the SNN would integrate the membrane potential, and would consume additional energy and latency [1]. In contrast, we merge the membrane potential and spike updates of the SNN for each time step with the analogous memory and hidden state updates of the LMUFormer for each input sample to develop our spiking LMUFormer model. Thus, our spiking LMUFormer requires the same number of time steps as the number of samples in the input sequence. However, spiking LMUFormer can not be easily accelerated on traditional GPU hardware, and thus incurs higher training and inference time compared to LMUFormer there. This phenomenon is also observed in traditional SNNs. Fortunately, spiking LMUFormer can be readily deployed on neuromorphic hardware, and thus can yield high energy efficiency there.
>
> [1] G. Datta, H. Deng, R. Aviles, Z. Liu and P. A. Beerel, "Bridging the Gap Between Spiking Neural Networks & LSTMs for Latency & Energy Efficiency," 2023 IEEE/ACM International Symposium on Low Power Electronics and Design (ISLPED), Vienna, Austria, 2023.

---

### Official Review · Reviewer_2bBb · 2023-11-01

**Soundness:** 3 good
**Presentation:** 3 good
**Contribution:** 3 good
**Rating:** 8
**Confidence:** 2

**Summary:**

This work combines Legendre Memory Units (LMU)  with convolutional patch embedding and convolutional channel mixer in its network design to improve accuracy. It also introduces a spiking version to further improve its inference efficiency. On speech recognition tasks, their results show a significant reduction in both model size and FLOPs compared to the SoTA AST model. It also achieves higher accuracy compared to other low-complexity SNN designs. It also shows it can achieve competitive accuracy with only two-thirds of the seq length due to its sequential processing capability.

**Strengths:**

- The addition of the convolutional patch embedding (w/ 1d conv) and convolutional channel mixer to LMU is well explained and justified.
- The model can be trained in parallel and has the streaming capability during inference. It does not need to wait until the entire input sequence is available, which can reduce the compute complexity and memory requirement.
-  This transformer design achieves competitive accuracy on a wide range of tasks compared to its counterparts with low compute complexity. It also demonstrates better classification accuracy in long token tasks compared to other linear-complexity attention designs.

**Weaknesses:**

- The results could be made stronger if the model size and compute complexity comparison can be added to every performance comparison table (Table 1,2,4).
- Ops represented in the paper for SNN can be a very optimistic proxy for latency performance. It is worth mentioning that SNNs might not be easily accelerated on off-the-shelf hardware like CPUs and GPUs. It might require specialized hardware to demstronate its advantage in latency and energy reduction.

**Questions:**

1. Is there a reason why Am(t) is changed to Am[t-1] from eqn(1) to eqn(2) for discretization?
2. From eqn (4), it seems the LMU module is composed of mainly matrix-vector operations, which is similar to RNN. Is it correct that we can also parallelize it across the time dimension? can you elaborate on how it can be solved in non iterative way to enable parallelized training?
3. I'm curious to know how its measured training and inference time compares to that of other linear-time transformers like Linformer on the off-the-shelf hardware.

---

> ### Author Response · Authors · 2023-11-17
> **Response to Reviewer 2bBb**
>
> Thanks for your positive comments and valuable suggestions to improve the quality of our work. Please see our response below.
>
> **Additional Comparison of Model Size, Compute Complexity, Training and Inference Time**
>
> We fully agree with you and we are happy to compare the model size and compute complexity comprehensively. However, since some studies do not open-source their model descriptions and there is some ambiguity regarding the exact model configuration, it is difficult to obtain accurate model sizes and FLOPs. Alternatively, we consider the accuracy, model size, peak memory, training time, and inference time, which allows us to compare the performance and efficiency of our LMUFormer with other models. As shown in the table below, our LMUFormer, similar to variants of the efficient Transformer Model, has a fast training speed.
>
>  | Model             | Text(4K) | Params. (M) | Peak memory (MB) | Training time (s) | Validation time (s) |
> |-------------------|----------|-------------|------------------|--------------------|---------------------|
> | Linear Trans.     | 65.90    | 0.174       | 644              | 5.91               | 2.37                |
> | Linformer         | 53.94    | 0.698       | 1180             | 8.69               | 2.46                |
> | Nystromformer     | 65.52    | 0.174       | 836              | 11.24              | 4.09                |
> | LMUFormer         | 68.27    | 0.174       | 738              | 6.29               | 2.46                |
> | Spiking LMUFormer | 65.80    | 0.175       | 928              | 12.02              | 5.66                |
> | pLMU              | 63.72    | 0.132       | 582              | 4.60               | 1.83                |
> | Spiking pLMU      | 60.84    | 0.132       | 684              | 8.68               | 3.75                |
>
> **Is there a reason why Am(t) is changed to Am[t-1] from eqn(1) to eqn(2) for discretization?**
>
> We are sorry that our omission has caused inconvenience to your understanding. The complete derivation steps using Euler method are as follows:
>
> \begin{align*}
>  \dot{\boldsymbol{m}}(t) &= \boldsymbol{A} \boldsymbol{m}(t) + \boldsymbol{B} \boldsymbol{u}(t) \\\\
> \frac{\boldsymbol{m}[t] - \boldsymbol{m}[t-1]}{\Delta t} &= \boldsymbol{A} \boldsymbol{m}[t] + \boldsymbol{B} \boldsymbol{u}[t] \\\\
> \boldsymbol{m}[t] &= \boldsymbol{m}[t-1] + {\Delta t}(\boldsymbol{A} \boldsymbol{m}[t] + \boldsymbol{B} \boldsymbol{u}[t])
> \end{align*}
> Since we assume $\Delta t$ is small enough, we can use use $\boldsymbol{m}[t-1]$ to approximate $\boldsymbol{m}[t]$. Therefore, the equation will be:
> \begin{align*}
> \boldsymbol{m}[t] &= (\boldsymbol{A} \cdot \Delta t + I)\boldsymbol{m}[t-1] + (\boldsymbol{B} \cdot \Delta t) \boldsymbol{u}[t] \\\\
> \boldsymbol{m}[t] &= \bar{\boldsymbol{A}} \boldsymbol{m}[t-1] + \bar{\boldsymbol{B}} \boldsymbol{u}[t]
> \end{align*}
>
> **Is it correct that we can also parallelize it across the time dimension? can you elaborate on how it can be solved in non iterative way to enable parallelized training?**
>
> Yes, you are right. As you can see, the only recurrent connection in the LMU cell is $\boldsymbol{m}[t] = \bar{\boldsymbol{A}} \boldsymbol{m}[t-1] + \bar{\boldsymbol{B}} \boldsymbol{u}[t]$, which calculates the memory of LMU cell at time step t, $\boldsymbol{m}[t]$, based on the memory at time step t-1, $\boldsymbol{m}[t-1]$, and the input of the LMU cell at time step t, $\boldsymbol{u}[t]$. $\bar{\boldsymbol{A}}$ and $\bar{\boldsymbol{B}}$ are the state-space metrics that are fixed and can be calculated before training.
>
> Since it is an LTI system, we can obtain $\boldsymbol{m}[t]$ in a non-iterative way: $\boldsymbol{m}[t] = \sum_{i=1}^t \bar{\boldsymbol{A}}^{t-i} \bar{\boldsymbol{B}} \boldsymbol{u}[i]$. Moreover, as shown in the following equation, we can get the memory at all time steps simultaneously by the matrix multiplication between a pre-calculated matrix $\boldsymbol{H} $ and the inputs matrix $\boldsymbol{U}$.
>
> $\begin{bmatrix}
>     \boldsymbol{m}[1] \\\\
>     \boldsymbol{m}[2] \\\\
>     \vdots \\\\
>     \boldsymbol{m}[t] \\\\
> \end{bmatrix} = [\bar{\boldsymbol{A}}^0\bar{\boldsymbol{B}},  \bar{\boldsymbol{A}}^1\bar{\boldsymbol{B}}, \cdots, \bar{\boldsymbol{A}}^{t-1}\bar{\boldsymbol{B}}]
> \begin{bmatrix}
>     \boldsymbol{u}[1] & \boldsymbol{u}[2] & \cdots & \boldsymbol{u}[t] \\\\
>      & \boldsymbol{u}[1] & \cdots & \boldsymbol{u}[t-1] \\\\
>      & & \ddots & \vdots \\\\
>      & & & \boldsymbol{u}[1] \\\\
> \end{bmatrix}= \boldsymbol{H} \boldsymbol{U}
> $.
>
> For RNN models, the parameter matrices $\bar{\boldsymbol{A}}$ and $\bar{\boldsymbol{B}} $ are not fixed as they are trainable, so the $\boldsymbol{H}$ needs to be recalculated at each forward. But they are still trained in parallel.
> For more details, you can check [1].
>
>
> [1] Narsimha Reddy Chilkuri and Chris Eliasmith. Parallelizing legendre memory unit training. In International Conference on Machine Learning, pp. 1898–1907. PMLR, 2021.

---

### Official Review · Reviewer_KEVY · 2023-11-01

**Soundness:** 2 fair
**Presentation:** 3 good
**Contribution:** 1 poor
**Rating:** 3
**Confidence:** 4

**Summary:**

This paper presents a sequential network that exploits Legendre Memory Units (LMU) module as its temporal computational core. It has been shown that the proposed network, which is called LMUFormer, can achieve a similar performance to transformers with less number of parameters and operations.

**Strengths:**

1) The accuracy performance of the proposed work is decent and the memory/operation reduction is significant.

2) The paper is also well-written and easy to read and understand.

**Weaknesses:**

1) The contribution of this work is rather limited as it relies on a previously proposed module (i.e., LMU). It is not also clear why the proposed network yields a better accuracy.

2) My main concern is the performance of such a network when pre-trained. The main advantage of transformers come from its pre-training stage which allows the model to perform well on downstream tasks during fine-tuning. To learn and store those pre-training data, transformers contains numerous parameters. As such, I am not supersized by the results reported in this paper since the transformer used for the comparison was not pre-trained and consequently it is expected to see a better performance from LMUFormer. What would be the performance of LMUFormer on well-known benchmarks such as GLUE? Can LMUFormer be pre-trained?

3) Lack of theoretical analysis and reasoning on the superior performance of LMUFormer.

**Questions:**

See my concerns listed as weaknesses.

---

> ### Author Response · Authors · 2023-11-17
> **Response to Reviewer KEVY**
>
> Thanks for your valuable comments and suggestions to improve the quality of our work. Please see our response below.
>
> **Why does LMUFormer yield superior performance?**
>
> 1. The original LMU model directly inputs the raw input data to the memory cell. However, previous research on vision transformers have shown that a patch embedding block consisting of several convolutional layers can improve the performance of vision tasks. Inspired by this idea, we propose 1D and 2D convolutional patch embedding blocks to extract more abstract features that are inputted to the memory cell of the LMU.
> 2. The original LMU model is a typical RNN which focuses on the information communication between different tokens. To augment this with the information communication between channels, we explicitly add some convolutional 1D layers, which we call the channel mixer. Therefore, both per-token features and per-channel features are sufficiently integrated in our model.
>
> Both these architectural optimizations explain the superior test accuracy of our LMUFormer, which is validated by the ablation studies presented in Table 5.
>
> **Pre-training of LMUFormer**
>
> We agree with you that pre-training significantly enhances the capabilities of Transformer models. However, the aim of this work is not to replace Transformers or develop models for a range of downstream tasks. Rather, our aim is to consider important streaming use-cases where Transformers exhibit limitations and to develop efficient models for these specific use-cases. Transformers need to wait for the entire sequence to start processing, which incur higher latency compared to our models. It is not very clear whether pre-training on a large text corpus that empower Transformers, can improve the performance of our models on streaming use-cases. Moreover, we are not aware of any large-scale streaming dataset, such as the BookCorpus for non-streaming NLP applications, that we can use to pre-train our models.
>
> Current pre-trained transformers, that can obtain high performance when fine-tuned on a range of downstream tasks, have high compute complexity, and can not be easily deployed on resource constrained edge devices, as our lightweight LMUFormer. Our novel spiking version of LMUFormer further reduces the compute complexity. However, we agree with you that, given a large-scale pre-trained dataset (and enough compute power), it may be possible for LMUFormer to scale up, and achieve high performance when fine-tuned on downstream tasks. We hypothesize this may not be possible by simply stacking LMUFormer blocks, and may require network architectural optimizations, which is an important research direction.
>
> These points have been added to the revision in Appendix A.2.

---

### Author Response · Authors · 2023-11-22
**Last day of discussion period**

Dear reviewers,

Thanks again for your insightful review and invaluable suggestions to improve the quality of our paper. As the author-reviewer discussion period is about to end (Nov 22, end-of-day AoE time), we kindly request you to take a look at our author response (and the updated version of the paper, which incorporates your suggestions) and let us know if your concerns are addressed or if you have any follow-up questions.

We would like to re-iterate the motivations and contributions of this work, along with some additional discussions, here for your reference.

We **aim to improve the accuracy-efficiency trade-off for streaming tasks** that we believe is crucial for edge AI. Since modern multi-head transformers are generally not suitable for these tasks (not only because of their large compute complexity, but also because they need to wait for the entire input sequence to start processing, which hinders real-time processing), we wonder **whether architectural modifications to a fully-sequential RNN can push its performance towards transformers, while still retaining its compute and latency benefits**. We build our work on LMUs since it is currently the state-of-the-art (SOTA) RNN for complex temporal tasks, and **propose a novel architecture, LMUFormer that augments the LMU with convolutional patch embedding and convolutional channel mixer**. This improves the representation capacity of the LMU, as validated from the significant accuracy increase in **community-standard streaming tasks** studied in this paper. Moreover, for extreme compute efficiency at the edge, we **present a novel spiking version of this LMUFormer, which introduces the benefit of states within the patch embedding and channel mixer modules**. The spiking architecture introduces high sparsity, and incurs accumulate-only operations which are significantly cheaper compared to multiply-and-accumulate operations required in the non-spiking counterpart.

Lastly, reviewer KEVY has asked an interesting question, whether our LMUFormer can be pre-trained like transformers. While indeed pre-training our LMUFormer and fine-tuning it on several streaming tasks would be useful, we are not aware of any large-scale pre-training benchmark for streaming applications that we study in this work. For example, pre-training our LMUFormer on large text corpus, would most likely not help in improving the performance on Google speech commands. Thus, creating such a benchmark is an interesting future direction. We are committed to further optimizing our LMUFormer network architecture to aid pre-training with such a benchmark (or modifying an existing benchmark) in the future.

Thanks,

Authors

---

### Meta-Review · Area_Chair_dQRk · 2023-12-04

**Metareview:**

The authors introduce the so-called LMUFormer model in this paper which adds convolutional patch embedding and convolutional channel mixer in Legendre Memory Units (LMU) to improve its modeling capability and most importantly to improve the accuracy-efficiency trade-off for streaming tasks. On top of that, the authors also introduce the spiking LMUFormer which promotes sparsity and therefore significantly boosts the compute efficiency.  The authors report competitive performance of LMUFormer on the Speech Commands V2 dataset compared to  transformer-based models with significant reduction in number of parameters and FLOPs.  When compared with SNN models, the spiking LMUFormer achieves the SOTA performance.  Overall, the work is well motivated with strong performance.  The topic under investigation is also important to the community.   The authors have meticulously addressed the questions and concerns raised by the reviewers in their rebuttal with additional experimental results and analysis.  Also, the authors have also discussed the limitation of the current work and possible future directions.  Although the technical novelty of the work is not overwhelmingly significant,  it may have its value to the machine learning community, especially edge computing.

**Justification For Why Not Higher Score:**

The technical novelty is not overwhelmingly significant.

**Justification For Why Not Lower Score:**

The performance of LMUFormer appears to be strong.

---

### Decision · Program_Chairs · 2024-01-16

Accept (poster)